# RUVBL1 in Clear-Cell Renal Cell Carcinoma: Unraveling Prognostic Significance and Correlation with HIF1A

**DOI:** 10.3390/cancers16071273

**Published:** 2024-03-25

**Authors:** Justyna Durślewicz, Aleksandra Maria Wybierała, Sara Szczepanek, Paulina Antosik, Damian Jaworski, Dariusz Grzanka

**Affiliations:** Department of Clinical Pathomorphology, Faculty of Medicine, Collegium Medicum in Bydgoszcz, Nicolaus Copernicus University in Torun, 85-094 Bydgoszcz, Poland; a.wybierala@cm.umk.pl (A.M.W.); s.szczepanek@cm.umk.pl (S.S.); paulina.antosik@cm.umk.pl (P.A.); damian.jaworski@cm.umk.pl (D.J.); d_grzanka@cm.umk.pl (D.G.)

**Keywords:** clear-cell renal cell carcinoma, RUVBL1, HIF1A, prognostic factor

## Abstract

**Simple Summary:**

This study delves into the roles of RUVBL1 and HIF1A in the development of clear-cell renal cell carcinoma (ccRCC) and investigates their clinical relevance as prognostic biomarkers. Utilizing TCGA data and an institutional tissue cohort, mRNA and protein expressions were analyzed. Elevated *RUVBL1* mRNA expression in ccRCC tissues was associated with advanced histological grade, T stage, lymph node metastasis, and clinical stage. High *RUVBL1* mRNA expression correlated with poorer overall survival, serving as an adverse prognostic factor. Similarly, *HIF1A* mRNA expression was significantly higher in ccRCC tissues and correlated with worse overall survival. Simultaneous evaluation of *RUVBL1* and *HIF1A* mRNA expression demonstrated enhanced prognostic capacity, surpassing individual markers. Immunohistochemical staining confirmed the upregulation of both RUVBL1 and HIF-1α proteins in ccRCC tissues, with high expression associated with shorter patient survival time. These findings emphasize RUVBL1 and HIF-1α as potential prognostic markers in ccRCC, suggesting avenues for translating these insights into clinical applications.

**Abstract:**

This study investigates the roles of RUVBL1 and HIF1A in ccRCC development and explores their clinical significance as prognostic biomarkers. mRNA and protein expressions were analyzed using TCGA data and an institutional tissue cohort, respectively. Correlations with clinicopathological parameters and patient outcomes were assessed. TCGA data revealed significantly elevated *RUVBL1* mRNA expression in ccRCC tissues, associated with advanced histological grade, T stage, lymph node metastasis, and clinical stage. High *RUVBL1* mRNA expression correlated with inferior overall survival and served as an adverse prognostic factor. Similarly, *HIF1A* mRNA expression was significantly higher in ccRCC tissues, correlating with worse overall survival and acting as an adverse prognostic factor for treatment outcomes. Simultaneous evaluation of *RUVBL1* and *HIF1A* mRNA expression demonstrated enhanced prognostic capacity, surpassing the predictive power of individual markers. Immunohistochemical staining confirmed substantial upregulation of both RUVBL1 and HIF-1α proteins in ccRCC tissues. Furthermore, high expression of both RUVBL1 and HIF-1α proteins was significantly associated with shorter patient survival time. Our findings underscore the significance of RUVBL1 and HIF-1α as potential prognostic markers in ccRCC, paving the way for further research to translate these insights into clinically relevant applications.

## 1. Introduction

Renal cell carcinoma (RCC) is one of the most common malignant urinary tumors. In 2020 globally, the estimated new cases and deaths amounted to 431,288 and 179,368, respectively [1]. Clear-cell renal cell carcinoma (ccRCC) represents the most prevalent histological subtype of RCC, accounting for approximately 85% of all renal cell carcinomas [2]. Within this group, 12–16% of individuals are diagnosed with distant metastases [3]. The five-year relative survival rate for those with metastatic ccRCC stands at 15.3% [4]. The advancements in ccRCC treatment, particularly with the introduction of immunotherapies, have significantly altered patient survival rates. These therapies, including immune checkpoint inhibitors, target the immune system to recognize and fight cancer cells more effectively. This shift towards immunotherapy, along with advancements in understanding the molecular basis of RCC, has improved survival rates but also highlighted the need for ongoing research to optimize treatment strategies [4,5].

In light of these findings, identifying reliable prognostic markers is of paramount importance in addressing the challenges posed by this malignant tumor. In this context, RuvB-like protein 1 (RUVBL1), emerges as a dynamic player [6,7,8]. RUVBL1, also known by synonyms such as Pontin52, Rvb1, TAP54α, and TIP49, is a highly conserved member of the AAA+ (ATPases associated with various cellular activities) superfamily of proteins [9,10,11]. With intrinsic ATPase and DNA helicase activities, *RUVBL1* emerges as a versatile player involved in transcription regulation, DNA damage repair, cell-cycle control, and stress adaptation [12]. Its dysregulation has been observed in multiple types of tumors, often correlating with patient prognosis. Under hypoxic conditions, the methylation process of *RUVBL1* by G9a/GLP methyltransferases becomes significant, enhancing *RUVBL1* activity in activating HIF1A by recruiting the p300 co-regulator to specific *HIF1A* target promoters. This leads to increased invasive and migratory properties of cells and the activation of target genes, such as Ets1 [6,8]. Earlier studies suggested conflicting roles for *RUVBL1* in *HIF1*-dependent transcription. However, research conducted by Perez-Perri et al. demonstrated that both *RUVBL1* and *Reptin* are essential for *HIF1* transcriptional activity. These findings shed light on the multifaceted role of *RUVBL1* in cellular responses to hypoxia and *HIF1*-dependent transcription [7].

This study aims to provide insights into the roles of RUVBL1 and HIF1A in the development of ccRCC and their potential clinical significance as prognostic biomarkers. mRNA expression was analyzed using publicly available TCGA data, while protein expression was assessed through immunohistochemistry in the institutional tissue cohort. Both expressions were then correlated with clinicopathological parameters and patient outcomes.

## 2. Materials and Methods

### 2.1. In Silico Analysis

*RUVBL1* and *HIF1A* mRNA levels were assessed in The Cancer Genome Atlas (TCGA) cohort and the Genotype-Tissue Expression (GTEx) cohort. RNA-sequencing transcriptome data for a cohort of 475 patients with ccRCC and 28 normal individuals were retrieved from the UCSC Xena Browser website (http://xena.ucsc.edu/, accessed on 11 April 2022) and normalized using DESeq2. The clinicopathological characteristics of the patients are provided in Appendix A. Gene-expression data were categorized into low and high expression levels based on the threshold determined using the cutp function of the Evaluate Cutpoints software [13]. The cut-off values for high and low *RUVBL1* and *HIF1A* expression were established at ≥10.97 and <10.97, and ≥12.95 and <12.95, respectively.

### 2.2. Patients and Tissue Material

Formalin-fixed paraffin-embedded (FFPE) specimens from 132 patients with ccRCC who underwent surgery at the Department of Urology and Andrology, Antoni Jurasz University Hospital No. 1 in Bydgoszcz, Poland, were initially screened for inclusion in the study. The histopathological diagnoses were independently confirmed by two pathologists. The clinical and pathological characteristics of the patients are detailed in Appendix A. This cohort was previously featured in our earlier study [14]. Due to the partial depletion of available material for analysis, a final inclusion of 99 samples was made from the initial pool of 132. Moreover, 34 adjacent non-tumor tissues were examined.

### 2.3. Immunohistochemistry

Immunohistochemicalstaining was conducted on tissue macroarrays constructed from representative tumor areas with a minimum of 80% tumor cells, as previously outlined [14]. RUVBL1 protein expression was detected using the BenchMark^®^ ULTRA automated slide-processing system by Roche Diagnostics/Ventana Medical Systems in Tucson, AZ, USA. Tissue sections were incubated with a primary rabbit polyclonal antibody against RUVBL1 (diluted 1:400, incubated for 32 min, catalog number: HPA019948, Sigma-Aldrich, St. Louis, MO, USA) and HIF-1α (diluted 1:200, incubated for 32 min, catalog number: MAB5382, Sigma-Aldrich, St. Louis, MO, USA). Antibody staining was visualized using the ultraView Universal DAB Detection Kit, also from Roche Diagnostics/Ventana Medical Systems in Tucson, AZ, USA. Additionally, quality control was performed using positive controls as recommended by the manufacturer and in accordance with the literature.

### 2.4. Immunostaining Evaluation

Our slides were subjected to digitization using a whole slide imaging scanner known as Roche Ventana DP 200 (Roche Diagnostics/Ventana Medical Systems in Tucson, AZ, USA), and they underwent evaluation by both a scientist and a pathologist. This evaluation was conducted using the modified Remmele–Stegner Index (IRS) scale. The final score, which falls within the range of 0 to 12, is calculated by multiplying two factors: the percentage of cells or areas displaying positive staining (ranging from 0 to 4) and the intensity of staining (ranging from 0 to 3). In the evaluation of RUVBL1, both nuclear and cytoplasmic staining were assessed. A positive outcome was assigned to cases exhibiting staining in both the nuclei and cytoplasm of cells. Conversely, a negative outcome indicated the absence of staining in both the nuclei and cytoplasm. In the evaluation of HIF-1α, emphasis was placed on nuclear staining. A positive outcome was ascribed to instances where staining manifested within the cell nuclei; conversely, a negative outcome denoted the lack of staining within the nuclei.

### 2.5. Statistical Analysis

Statistical analysis was performed using SPSS software version 26.0 (IBM Corporation, Armonk, NY, USA) and GraphPad Prism version v10.01 (GraphPad Software, San Diego, CA, USA). The normality of the data distribution was assessed with the Shapiro–Wilk test. Continuous variables were compared using the Mann–Whitney test, while categorical variables were analyzed with the chi-square test or Fisher’s exact test. Spearman’s correlation coefficient was used to assess the correlations between the expression of RUVBL1 and HIF1A. Survival data were analyzed using the Kaplan–Meier method, and differences between survival curves were evaluated with the log-rank test. Univariate and multivariate survival analyses were conducted using the Cox proportional hazards regression model to estimate hazard ratios (HR) along with 95% confidence intervals (CI). Multivariate Cox proportional hazard analyses for both our cohort and the TCGA cohort were adjusted for gender (male vs. female), age (≤60 years vs. >60 years for the TCGA cohort; ≤64 years vs. >64 years for our cohort), grade (G1, G2 vs. G3), and AJCC pathologic stage (stage I, II vs. stage III, IV). A significance level of *p* < 0.05 was considered statistically significant.

## 3. Results

### 3.1. RUVBL1 and HIF1A mRNA Expression in Tumor and Normal Adjacent Tissue Derived from Public Datasets: Clinicopathological Associations

As depicted in Figure 1, the ccRCC samples exhibited significantly elevated expression of *RUVBL1* compared to the control tissue (*p* < 0.0001). High expression of *RUVBL1* mRNA was observed in 151 cases (31.79%). We noted that the frequency of high *RUVBL1* mRNA expression increased with tumor grades (*p* = 0.0337), pT status (*p* = 0.0009), and TNM stage (*p* = 0.0005). Elevated mRNA levels of *RUVBL1* were also more frequently detected in patients with lymph node metastases compared to those without cancer cells in the lymph nodes (*p* = 0.001, Table 1).

The ccRCC samples exhibited significantly elevated *HIF1A* expression compared to the control tissue (*p* = 0.008). High mRNA expression of *HIF1A* was observed in 227 (47.79%) cases, while low expression was found in 248 (52.21%) cases. No significant associations were observed between the *HIF1A* expression level and the clinical–pathological data (Table 1).

### 3.2. Prognostic Value of RUVBL1 and HIF1A mRNA Expression from TCGA in Predicting the Overall Survival of ccRCC Patients

The Kaplan–Meier survival analysis within the TCGA cohort unveiled that the high-expression group of *RUVBL1* exhibited a significantly shorter median overall survival (OS) in comparison to the low-expression group (1980 days vs. undefined days; *p* < 0.0001; Figure 2). A univariate analysis indicated a significant association between positive *RUVBL1* expression and a worsened survival prognosis (HR 1.82, 95% CI 1.33–2.50, *p* < 0.0001; Table 2). In the multivariate analysis, after adjusting for clinical factors, *RUVBL1* retained its status as an unfavorable prognostic factor for OS (HR 1.56, 95% CI 1.13–2.14, *p* = 0.006; Table 2).

The Kaplan–Meier survival analysis in the TCGA cohort revealed that the high-expression group of *HIF1A* had a significantly shorter median OS in comparison to the low-expression group (2105 days vs. undefined days; *p* < 0.0001; Figure 2). A univariate analysis demonstrated that positive *HIF1A* expression was significantly associated with a worse survival prognosis (HR 1.48, 95% CI 1.09–2.03, *p* = 0.014; Table 2). In the multivariate analysis, after adjusting for clinical factors, *HIF1A* remained an adverse prognostic factor for OS (HR 1.60, 95% CI 1.17–2.21, *p* = 0.004; Table 2).

Having established the individual factors as independent prognostic markers, we proceeded to investigate the combined impact of their expression on OS in the TCGA cohort. A weak positive and significant association was found between *RUVBL1* and *HIF1A* expression (*p* < 0.0001, Spearman coefficient r = 0.226). Kaplan-–Meier analysis disclosed a statistically significant reduction in OS among patients exhibiting concurrent high expressions of *RUVBL1* and *HIF1A*, with a median OS of 1912 days. This was in contrast to patients displaying low expressions of both *RUVBL1* and *HIF1A*, where the median OS remained undefined. In a multivariate analysis, after adjusting for clinical factors, both *RUVBL1* and *HIF1A* remained unfavorable prognostic factors for OS (*RUVBL1*: HR 1.43, 95% CI 1.03–1.98, *p* = 0.03; *HIF1A*: HR 1.50, 95% CI 1.08–2.08, *p* = 0.01; Table 2).

### 3.3. Immunoexpression of RUVBL1 and HIF-1α in ccRCC and Normal Adjacent Tissue: Clinicopathological Associations

To investigate and validate a hypothesis based on preliminary data extracted from publicly accessible databases, we conducted immunohistochemical staining. The expression of RUVBL1 and HIF-1α proteins in our cohort was assessed using immunohistochemistry in ccRCC tissue samples and adjacent non-tumor tissues. Representative microphotographs are shown in Figure 3.

High immunoreactivity of RUVBL1 was found in 59 (59.60%) ccRCC cases, whereas the remaining 40 (40.40%) demonstrated low expression. The ccRCC samples demonstrated a significantly higher level of RUVBL1 expression in comparison to adjacent normal tissue (*p* = 0.0336, Figure 4). No significant associations were observed between the RUVBL1 expression level and clinical–pathological data (Table 3). Among the tumor tissues, 40 samples (40.40%) exhibited nuclear expression of HIF-1α, while 59 samples (59.60%) showed its absence. The ccRCC samples exhibited a significantly elevated level of HIF-1α expression compared to the adjacent normal tissue (*p* = 0.0067, Figure 4). No noteworthy correlations were identified between the expression level of HIF-1α and the clinical–pathological data (Table 3).

### 3.4. Prognostic Value of RUVBL1 and HIF-1α Immunoexpression in Predicting the Overall Survival of ccRCC Patients

The Kaplan–Meier survival curves indicated a significantly poorer OS (*p* = 0.004) in ccRCC patients with a high RUVBL1 protein expression compared to those with a low expression (median OS of 867 and 1825 days, respectively; HR 1.86, 95% CI 1.21–2.87, *p* < 0.01; Figure 5, Table 4). In the multivariate analysis, after adjusting for gender (females had worse overall survival rates), age, grade, and N status, high RUVBL1 protein expression was identified as an independent adverse prognostic factor for OS (HR 2.00, 95% CI 1.27–3.13, *p* < 0.01; Table 4).

The Kaplan–Meier survival curves revealed a statistically significant decrease in OS (*p* = 0.038) among ccRCC patients exhibiting a high HIF-1α protein expression compared to those with a low expression (median OS of 1086 and 1844 days, respectively; HR 1.55, 95% CI 1.02–2.35, *p* = 0.04; Figure 5, Table 4). However, in the multivariate analysis, after adjusting for relevant factors, high HIF-1α protein expression did not emerge as an independent adverse prognostic factor for OS.

A weak positive and significant association was found between RUVBL1 and HIF-1αexpression (*p* = 0.009, Spearman coefficient r = 0.258). After establishing the individual factors as independent prognostic markers, we proceeded to examine the collective impact of their expression on OS in our cohort. A Kaplan–Meier analysis revealed no statistically significant difference in OS. In the multivariate analysis, after adjusting for clinical factors, only RUVBL1 remained an unfavorable prognostic factor for OS (HR 2.06, 95% CI 1.27–3.34, *p* < 0.01; Table 4).

## 4. Discussion

Our analysis, based on data from TCGA, revealed a significant elevation in *RUVBL1* mRNA expression in ccRCC tissues compared to control tissues. Notably, the heightened levels of *RUVBL1* mRNA expression were intricately associated with adverse clinicopathological features. High *RUVBL1* mRNA expression was correlated with a higher histological grade, advanced T stage, lymph node metastasis, and advanced clinical stage. Furthermore, the influence of *RUVBL1*-high mRNA expression levels on patient outcomes was evident, with a clear association with inferior overall survival rates. We also identified high *RUVBL1* mRNA expression as a significant adverse prognostic factor for ccRCC patient outcomes. These findings strongly suggest that the levels of *RUVBL1* mRNA expression could serve as a valuable indicator for predicting the clinical course and outcome in ccRCC patients. Subsequently, we conducted immunohistochemical staining to examine the expression of the *RUVBL1* protein within our cohort, focusing on tissue samples from ccRCC and adjacent non-tumor tissues. Our analyses, in line with the findings in the TCGA cohort, revealed a substantial upregulation in the protein expression of RUVBL1 in ccRCC tissues when compared to the corresponding control tissues. These discoveries are consistent with previous studies, which also noted the elevated expression of RUVBL1 in other cancer types, such as oral squamous cell carcinoma [15], osteosarcoma [16], salivary gland cancer [17], lung cancer [18,19], hepatocellular carcinoma [20], colorectal carcinoma [21,22], and glioma [23]. In the study conducted by Zhang et al., they also observed strong cytoplasmic and nuclear staining of RUVBL1 in ccRCC tissues, whereas non-tumor tissues exhibited either no expression of RUVBL1 or predominantly moderate nuclear expression [24]. It is worth noting that the authors performed a Kaplan–Meier analysis, revealing that high cytoplasmic expression of RUVBL1 is associated with poorer prognosis [24]. This finding suggests the potential utility of RUVBL1 as a prognostic biomarker in RCC. However, our study centered on the simultaneous analysis of nuclear and cytoplasmic expression. Our findings bolstered the idea that both forms of expression are intricately associated with a shorter survival time, offering additional insights into the multifaceted role of RUVBL1 in the context of ccRCC. The association between high expression levels and a shorter survival time has also been observed in studies conducted by other authors [17,19,23,24,25,26]. The cumulative evidence robustly underscores the prospective clinical significance of RUVBL1 as a prognostic marker in ccRCC, underscoring its importance in forecasting the overall survival of patients.

Our analyses, utilizing data from TCGA, revealed a substantial increase in the mRNA expression of *HIF1A* within ccRCC tissues in comparison to control tissues. Furthermore, high levels of *HIF1A* mRNA expression were associated with worse overall survival in patients. We also observed that a high mRNA expression of *HIF1A* was a significant adverse prognostic factor for the treatment outcomes of ccRCC patients. Following this, we performed immunohistochemical staining to assess the presence of HIF-1α proteins within our cohort, specifically examining tissue samples from both ccRCC and adjacent non-tumor tissues. Consistent with the results obtained from the TCGA cohort, our analyses demonstrated a significant elevation in the protein expression of HIF-1α in ccRCC tissues compared to the corresponding control tissues. Our findings align with the previous research indicating elevated HIF-1α expression across various cancer types, including renal cancer [27,28], glioblastoma [29], non-small-cell lung cancer [30], thyroid carcinoma [31], bladder cancer [32], cervical cancer [33], endometrial carcinoma [34], gastric cancer [35], and rectal cancer [36,37]. Notably, this increased expression was associated with an unfavorable prognosis for patients with astrocytomas [38], laryngeal carcinoma [39], endometrial carcinoma [34], gastric cancer [35], cervical cancer [33], bladder cancer [32], and rectal cancer [36,37]. Intriguingly, Lidgren et al. observed in their study that patients with ccRCC and high levels of HIF-1a tended to exhibit a more favorable prognosis, contradicting our own results [40]. It is important to highlight that the discrepancy may be attributed to the fact that Lidgren and colleagues analyzed cytoplasmic staining, while our study focused on the nuclear staining pattern. This emphasizes the need to recognize the distinct subcellular localization of proteins when assessing their prognostic value, as it can significantly influence the observed outcomes.

Ultimately, taking into consideration the interplay between *RUVBL1* and *HIF1A*, we initiated an investigation aimed at evaluating the cumulative impact of the co-expression of these genes on the OS of patients diagnosed with ccRCC [7,8]. The application of Kaplan–Meier survival analysis unveiled a compelling revelation: patients with tumors characterized by elevated mRNA expression levels of both *RUVBL1* and *HIF1A* experienced the least favorable OS outcomes, while a low expression of *RUVBL1* and *HIF1A* was associated with extended patient survival. It is noteworthy that the simultaneous expression of these two molecular markers demonstrated an enhanced prognostic capacity in predicting patient survival compared to evaluating each marker alone. This suggests that assessing the simultaneous mRNA expression of *RUVBL1* and *HIF1A* provides a more powerful prognostic tool compared to evaluating each marker separately. However, when we scrutinized the protein expression results for our own cohort, we did not observe similar outcomes. This suggests a potential discrepancy between mRNA and protein expression patterns and emphasizes the complexity of interpreting biomarker data in the context of ccRCC prognosis. The final conclusions of the study indicate the potential value of assessing the concurrent expression of *RUVBL1* and *HIF1A* as a powerful prognostic tool. However, they also highlight the necessity of considering the complexity of biological regulatory mechanisms between mRNA and protein. This opens the door for further research to precisely understand these relationships in the context of renal cell carcinoma prognosis.

Recognizing a limitation in our study related to the modest cohort size, the limited number of patients may impact the ability to draw statistically robust conclusions. This underscores the need for caution in interpreting the results and emphasizes the importance of additional research with larger cohorts to authenticate and strengthen our findings. However, in light of the obtained results and data available in the literature, the role of RUVBL1 in ccRCC may have therapeutic implications that could significantly impact future treatment strategies for this condition. A pivotal aspect is targeted therapy, with the recognition of RUVBL1 as a key player in the activation of HIF-1α and HIF-1-dependent transcription offering new avenues for molecularly targeted interventions. The precision of drug targeting to these molecular mechanisms is likely crucial for treatment effectiveness. Furthermore, investigations into the methylation of RUVBL1 under hypoxic conditions suggest that inhibiting this methylation process could be a promising therapeutic target. Such inhibition might influence the activity of RUVBL1 and its role in regulating HIF-1α.

## 5. Conclusions

In conclusion, our study provides valuable insights into the intricate involvement of RUVBL1 and its interaction with HIF1A, establishing a foundation for future research endeavors with larger cohorts. The multifaceted role of RUVBL1 in ccRCC unraveled in our analysis paves the way for the development of targeted therapeutic approaches, offering promising avenues for the effective management of this challenging malignancy.

## Figures and Tables

**Figure 1 cancers-16-01273-f001:**
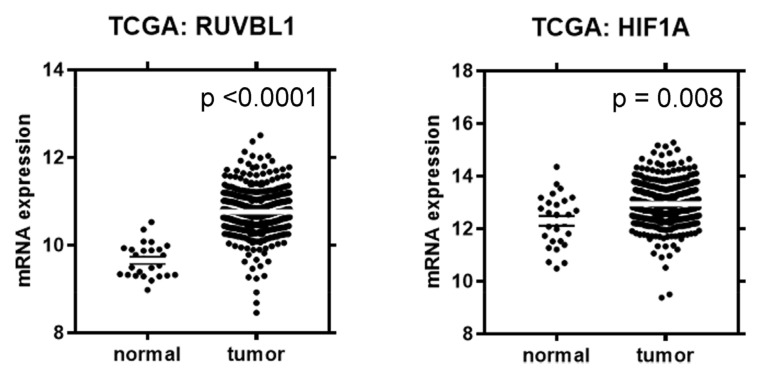
Comparison of *RUVBL1* and *HIF1A* mRNA expression levels in ccRCC and normal tissues.

**Figure 2 cancers-16-01273-f002:**
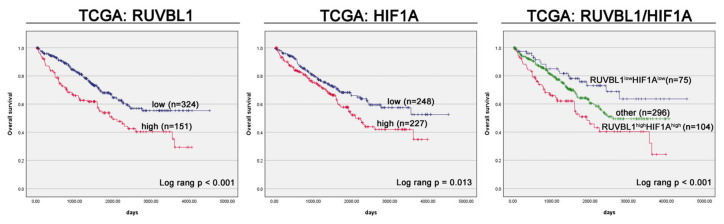
Kaplan–Meier curves illustrating the overall survival of ccRCC patients, stratified by *RUVBL1*, *HIF1A*, and the combination of *RUVBL1/HIF1A* mRNA expression from the TCGA cohort.

**Figure 3 cancers-16-01273-f003:**
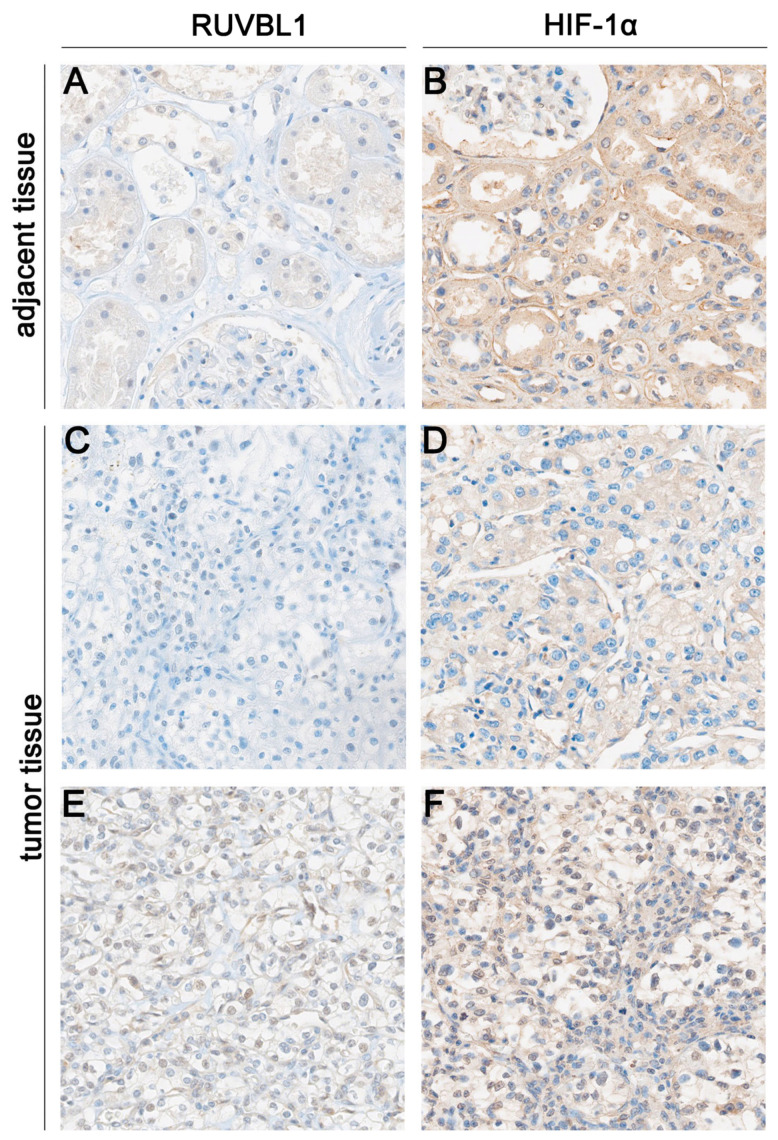
Representative microphotographs illustrating immunohistochemical expression of RUVBL1 (**A**,**C**,**E**) and HIF-1α (**B**,**D**,**F**) in ccRCC. RUVBL1 staining in adjacent normal tissues (**A**); absence of RUVBL1 staining (**C**); presence of RUVBL1 staining in ccRCC (**E**); HIF-1α staining in adjacent normal tissues (**B**); absence of HIF-1α staining (**D**); presence of HIF-1α staining in ccRCC (**F**); The original magnification is 20×.

**Figure 4 cancers-16-01273-f004:**
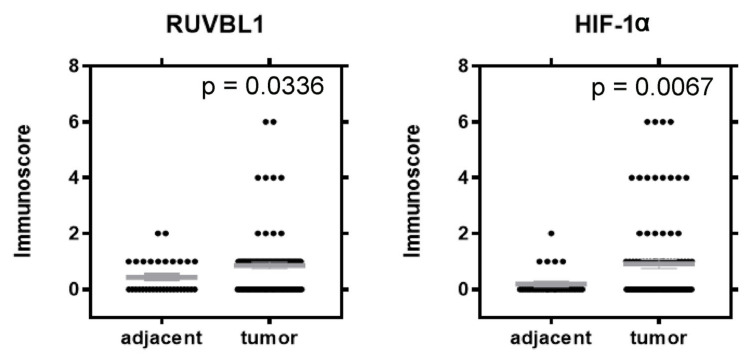
Comparison of RUVBL1 and HIF-1α protein expression levels in ccRCC and adjacent non-tumor tissues.

**Figure 5 cancers-16-01273-f005:**
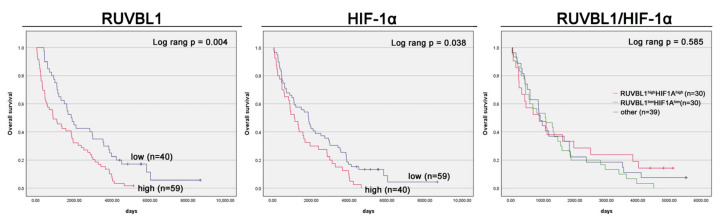
Kaplan–Meier curves depicting the overall survival of ccRCC patients, stratified by RUVBL1, HIF-1α, and the combination of RUVBL1/HIF-1α protein expression.

**Table 1 cancers-16-01273-t001:** Association between *RUVBL1* and *HIF1A* and clinicopathological features in mRNA cohort of ccRCC patients.

		*RUVBL1*			*HIF1A*		
Variables	Number (%)	HIGH	LOW	*p*-Value	HIGH	LOW	*p*-Value
		*n* = 151	*n* = 324		*n* = 227	*n* = 248	
Gender							
Females	163 (34.32)	46 (28.22)	117 (71.78)	0.2541	70 (42.94)	93 (57.06)	0.1467
Males	312 (65.68)	105 (33.65)	207 (66.35)		157 (50.32)	155 (49.68)	
Age							
≤60	239 (50.32)	79 (33.05)	160 (66.95)	0.5562	120 (50.21)	119 (49.79)	0.3127
>60	236 (49.68)	72 (30.51)	164 (69.49)		107 (45.34)	129 (54.66)	
Grade							
G1	11 (2.32)	1 (9.09)	10 (90.91)	0.0337	5 (45.45)	6 (54.55)	0.6386
G2	203 (42.74)	56 (27.59)	147 (72.41)		91 (44.83)	112 (55.17)	
G3	189 (39.79)	63 (33.33)	126 (66.67)		97 (51.32)	92 (48.68)	
G4	72 (15.16)	31 (43.06)	41 (56.94)		34 (47.22)	38 (52.78)	
pT status							
T1	237 (49.89)	56 (23.63)	181 (76.37)	0.0009	113 (47.68)	124 (52.32)	0.1763
T2	61 (12.84)	23 (37.70)	38 (62.30)		31 (50.82)	30 (49.18)	
T3	167 (35.16)	66 (39.52)	101 (60.48)		75 (44.91)	92 (55.09)	
T4	10 (2.11)	6 (60.00)	4 (40.00)		8 (80.00)	2 (20.00)	
pN status							
Nx	235 (49.47)						
N0	225 (47.37)	67 (29.78)	158 (70.22)	0.001	109 (48.44)	116 (51.56)	0.7932
N1	15 (3.16)	11 (73.33)	4 (26.67)		8 (53.33)	7 (46.67)	
Stage							
I	234 (49.26)	56 (23.93)	178 (76.07)	0.0005	112 (47.86)	122 (52.14)	0.1433
II	50 (10.53)	16 (32.00)	34 (68.00)		26 (52.00)	24 (48.00)	
III	119 (25.05)	44 (36.97)	75 (63.03)		48 (40.34)	71 (59.66)	
IV	72 (15.16)	35 (48.61)	37 (51.39)		41 (56.94)	31 (43.06)	

**Table 2 cancers-16-01273-t002:** Univariate and multivariate Cox proportional hazards models for OS in the TCGA cohort of ccRCC patients.

	Univariate Analysis				Multivariate Analysis *RUVBL1*				Multivariate Analysis *HIF1A*				Multivariate Analysis *RUVBL1* and *HIF1A*			
Variable	HR	95% CI		*p*-Value	HR	95% CI		*p*-Value	HR	95% CI		*p*-Value	HR	95% CI		*p*-Value
RUVBL1	1.82	1.33	2.50	<0.001	1.57	1.14	2.16	0.01	-	-	-	-	1.43	1.03	1.98	0.03
HIF1A	1.48	1.08	2.03	0.01	-	-	-	-	1.60	1.17	2.21	<0.01	1.50	1.08	2.08	0.01
gender	0.95	0.68	1.31	0.74	0.89	0.64	1.24	0.49	0.86	0.62	1.20	0.38	0.87	0.63	1.22	0.43
age	1.01	1.00	1.03	0.08	0.94	0.68	1.29	0.70	0.96	0.70	1.32	0.81	1.01	0.99	1.02	0.23
grade	1.36	0.98	1.87	0.06	1.16	0.84	1.61	0.37	1.17	0.84	1.61	0.35	1.15	0.83	1.59	0.40
pT	3.19	2.31	4.39	<0.001	-	-	-	-	-	-	-	-	-	-	-	-
pN	3.65	1.93	6.90	<0.001	-	-	-	-	-	-	-	-	-	-	-	-
TNM stage	3.61	2.59	5.02	<0.001	3.40	2.43	4.75	<0.001	3.66	2.62	5.11	<0.001	3.43	2.45	4.80	<0.001

**Table 3 cancers-16-01273-t003:** Association between RUVBL1 and HIF-1α and clinicopathological features in our cohort of ccRCC patients.

		RUVBL1			HIF-1α		
Variables	Number (%)	HIGH	LOW	*p*-Value	HIGH	LOW	*p*-Value
		*n* = 59	*n* = 40		*n* = 40	*n* = 59	
Gender							
Females	31 (31.31)	20 (64.52)	11 (35.48)	0.6592	13 (41.94)	18 (58.06)	0.8294
Males	68 (68.69)	39 (57.35)	29 (42.65)		27 (39.71)	41 (60.29)	
Age							
≤64	58 (58.59)	31 (53.45)	27 (46.55)	0.1519	21 (36.21)	37 (63.79)	0.4059
>64	41 (41.41)	28 (68.29)	13 (31.71)		19 (46.34)	22 (53.66)	
Grade							
G1	25 (25.25)	16 (64.00)	9 (36.00)	0.6224	9 (36.00)	16 (64.00)	0.3977
G2	64 (64.65)	36 (56.25)	28 (43.75)		25 (39.06)	39 (60.94)	
G3	10 (10.10)	7 (70.00)	3 (30.00)		6 (60.00)	4 (40.00)	
pT status							
Tx	1						
T1	28 (28.57)	15 (53.57)	13 (46.43)	0.2193	12 (42.86)	16 (57.14)	0.8554
T2	28 (28.57)	14 (50.00)	14 (50.00)		10 (35.71)	18 (64.29)	
T3-T4	42 (42.86)	29 (69.05)	13 (30.95)		17 (69.05)	25 (30.95)	
pN status							
N0	92 (92.93)	54 (58.70)	38 (41.30)	0.6979	36 (39.13)	56 (60.87)	0.4356
N1	7 (7.07)	5 (71.43)	2 (28.57)		4 (57.14)	3 (42.86)	

**Table 4 cancers-16-01273-t004:** Univariate and multivariate analyses of prognostic factors by the Cox proportional hazard model in our cohort of ccRCC patients.

	Univariate Analysis				Multivariate Analysis: RUVBL1				Multivariate Analysis: HIF-1α				Multivariate Analysis: RUVBL1 and HIF-1α			
Variable	HR	95% CI		*p*-Value	HR	95% CI		*p*-Value	HR	95% CI		*p*-Value	HR	95% CI		*p*-Value
RUVBL1	1.86	1.21	2.87	<0.01	2.00	1.27	3.13	<0.01	-	-	-	-	2.06	1.27	3.34	<0.01
HIF-1α	1.55	1.02	2.35	0.04	-	-	-	-	1.33	0.86	2.05	0.20	1.03	0.64	1.64	0.91
gender	0.56	0.36	0.87	0.01	0.62	0.39	0.97	0.04	0.58	0.37	0.91	0.02	0.63	0.40	0.99	0.05
age	1.62	1.07	2.47	0.02	1.43	0.92	2.22	0.11	1.29	0.82	2.01	0.27	1.40	0.90	2.20	0.14
grade	2.90	1.48	5.68	<0.01	3.27	1.62	6.62	<0.01	2.65	1.32	5.30	0.01	3.42	1.67	6.97	0.00
pT status	1.14	0.75	1.73	0.54	-	-	-	-	-	-	-	-	-	-	-	-
N status	3.77	1.68	8.47	<0.01	3.07	1.35	7.01	0.01	3.20	1.40	7.31	0.01	3.12	1.35	7.17	0.01

## Data Availability

The datasets generated and analyzed during the current study can be obtained from the corresponding author upon reasonable request.

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
