# Peer review of "RUVBL1 in Clear-Cell Renal Cell Carcinoma: Unraveling Prognostic Significance and Correlation with HIF1A"

_cancers, 2024, doi:10.3390/cancers16071273_

Round 1

Reviewer 1 Report

Comments and Suggestions for Authors

1)     General comments

The authors have evaluated RUVBL1 and HIF1A mRNA expression using TCGA data and shown that it correlates with prognosis in patients with clear cell renal cell carcinoma (CCRCC). Furthermore, the authors have validated the expression of RUVBL1 and HIF1A by immunostaining from their own cohort data and shown that their expression could be a prognostic factor in CCRCC patients. Observations about the expression of RUVBL1 and HIF1A, and prognosis in CCRCC patients deserve an interest. However, there are several points that need clarification.

2)     Specific comments for revision

Major comments

a)      The author describes the prognosis of patients with metastatic RCC in the "Introduction" section. However, the treatment of metastatic RCC has now been dramatically extended with the advent of Immune-oncology therapy. The reports cited by the authors as REFERENCES appear to be too old. The authors' description may give a false perception to the reader and should be corrected.

b)     In the "Materials and Methods" section, the authors indicate cut-off values for RUVBL1 and HIF1A to separate high and low expression. The authors should state how the cut-off values were set.

c)      The authors assessed the expression of RUVBL1 in immunostaining, both nuclear and cytoplasmic. What are the differences between the nucleus and cytoplasm of RUVBL1? Please clarify.

Miner comments

a)     In TCGA: HIF1A in Figure 2, please correct low group to (n=248).

b)     In Table 3, please correct “T3 i T4”.

c)     Although gender is shown as an independent prognostic factor in the univariate and multivariate analysis in Table 4, the authors should state in the text which gender had a worse prognosis.

Reviewer 2 Report

Comments and Suggestions for Authors

Submitted manuscript presents study investigating prognostic significance of RUVBL1 and HIF1A expression levels in clear cell renal cell carcinoma (ccRCC) patients. The authors used in silico analysis TCGA datasets and immunohistochemistry (IHC) analysis of formalin-fixed paraffin-embedded specimens. Their findings suggest that simultaneous evaluation of RUVBL1 and HIF1A expression levels in cancerous tissues can be considered as useful prognostic biomarker in patients with ccRCC. Submitted study provides novel data, combining well-known prognostic/predictive biomarker HIF1A with one of its cofactor, RUVBL1. The results are clearly presented and the findings support conclusions. However, there are some remarks that must be addressed in the revised version of the article before it could be considered for publication.

1.      Specify the type of positive controls used for IHC. Did the authors carry out any negative controls?

2.      It is unclear which cells were analyzed for their RUVBL1 and HIF1A immunoreactivity, especially in sections of non-cancerous, normal renal tissue.

3.      Why kidney medulla is shown in Figure 3? Which cells of the unchanged renal tissue were investigated for RUVBL1 and HIF1A immunoexpression? Since there is a consensus on the origin of ccRCC cells, immunoreactivity in epithelial cells of proximal convoluted tubule shall be evaluated.

4.      Figure 3 Legend must be carefully revised since much more comprehensive description of IHC results and localization of immunoreactive cells is necessary. Authors shall add letters to images and references in the main text body and/or figure legend.

5.      The presented tables with Cox regression suggest that the multivariate analyses were carried out separately. When building a model for multivariate Cox regression both RUVBL1 and HIF1A shall be included in a model (in order to support or invalidate them as independent prognostic factors) Did authors carry out such an analysis?

6.      The discussion shall be much more concise and focused on authors’ findings; e.g. lines 249-273 shall be transferred (after shortening) to the introduction.

Round 2

Reviewer 1 Report

Comments and Suggestions for Authors

The author has corrected the manuscript sufficiently.

I have no specific new comments to modify.